# A Transcriptomics Analysis of the Regulation of Lens Fiber Cell Differentiation in the Absence of FGFRs and PTEN

**DOI:** 10.3390/cells13141222

**Published:** 2024-07-19

**Authors:** Anil Upreti, Stephanie L. Padula, Jacob M. Weaver, Brad D. Wagner, Allison M. Kneller, Anthony L. Petulla, Salil A. Lachke, Michael L. Robinson

**Affiliations:** 1Cell, Molecular and Structural Biology Program, Miami University, Oxford, OH 45056, USA; upretia@miamioh.edu (A.U.); stephanielpadula@gmail.com (S.L.P.); weaverj9@miamioh.edu (J.M.W.); 2Department of Biology and Center for Visual Sciences, Miami University, Oxford, OH 45056, USA; wagnerbd@miamioh.edu (B.D.W.); knelleam@miamioh.edu (A.M.K.); antpetulla@gmail.com (A.L.P.); 3Department of Biological Sciences, University of Delaware, Newark, DE 19716, USA; salil@udel.edu; 4Center for Bioinformatics and Computational Biology, University of Delaware, Newark, DE 19716, USA

**Keywords:** lens, lens explants, FGFRs, PDGFRs, PTEN, RNA-seq, inflammation, EMT

## Abstract

Adding 50% vitreous humor to the media surrounding lens explants induces fiber cell differentiation and a significant immune/inflammatory response. While *Fgfr* loss blocks differentiation in lens epithelial explants, this blockage is partially reversed by deleting *Pten*. To investigate the functions of the *Fgfrs* and *Pten* during lens fiber cell differentiation, we utilized a lens epithelial explant system and conducted RNA sequencing on vitreous humor-exposed explants lacking *Fgfrs*, or *Pten* or both *Fgfrs* and *Pten.* We found that *Fgfr* loss impairs both vitreous-induced differentiation and inflammation while the additional loss of *Pten* restores these responses. Furthermore, transcriptomic analysis suggested that PDGFR-signaling in FGFR-deficient explants is required to mediate the rescue of vitreous-induced fiber differentiation in explants lacking both *Fgfrs* and *Pten.* The blockage of β-crystallin induction in explants lacking both *Fgfrs* and *Pten* in the presence of a PDGFR inhibitor supports this hypothesis. Our findings demonstrate that a wide array of genes associated with fiber cell differentiation are downstream of FGFR-signaling and that the vitreous-induced immune responses also depend on FGFR-signaling. Our data also demonstrate that many of the vitreous-induced gene-expression changes in *Fgfr*-deficient explants are rescued in explants lacking both *Fgfrs* and *Pten*.

## 1. Introduction

Lens fiber cell differentiation requires the activation of both ERK- and AKT-signaling cascades, which can be induced by multiple receptor tyrosine kinases (RTKs). Although FGF is the only growth factor capable of independently inducing lens fiber cell differentiation in lens epithelial explants, other RTK-activating growth factors, namely insulin-like growth factor 1 (IGF1), epidermal growth factor (EGF), vascular endothelial growth factor (VEGF) and platelet-derived growth factor (PDGF), contribute to lens development [1,2]. The lens epithelium expresses receptors for all of these growth factors. The mouse lens expresses *Fgfr1*, *Fgfr2* [3] and *Fgfr3* [4], with the expression of *Fgfr1* and *Fgfr3* increasing significantly as fiber cell differentiation commences [5]. In contrast, the platelet-derived growth factor receptor alpha (*Pdgfra*) is restricted to the lens epithelium, with expression levels falling sharply upon fiber cell differentiation [6]. 

The Overbeek lab created various transgenic mouse lines that overexpressed RTK ligands in the lens, driven by the αA-crystallin promoter. The ligands included TGFα [7], EGF [8], IGF1 [9], VEGF [10], PDGF-A [11] and multiple different FGFs. Of these transgenic mice, only FGFs were capable of inducing a fiber cell-differentiation response in the lens epithelium in vivo [12,13,14]. The lens epithelium of transgenic mice overexpressing PDGF-A showed some early features of fiber cell differentiation, namely cell elongation and β-crystallin expression, but these lens cells failed to exit the cell cycle and express characteristics of mature fiber cells. Among the RTKs expressed in the lens epithelium, FGFRs uniquely interact with FRS2, a protein that docks with FGFRs and mediates sustained ERK1/2 activation [15]. FRS2 also interacts with the TRK family of RTKs. Although TRKC is not normally expressed in lens epithelial cells, transgenic mice expressing TRKC in the lens epithelium undergo fiber cell differentiation in vivo in response to the TRKC ligand, NT-3 [16]. 

Of the RTKs expressed in the lens, only the loss of FGFRs blocks lens fiber cell differentiation in vivo [17]. The loss of *Fgfr1, Fgfr2* and *Fgfr3* is required to block fiber cell differentiation, but this effect can only be observed if these receptors are removed after the lens vesicle stage (E10.5), because loss of *Fgfr1* and *Fgfr2* alone at the lens placode stage (E9.0) causes such severe apoptosis that the lens degenerates prior to fiber cell formation [17,18]. Given the substantial increase in lens apoptosis caused by FGFR loss [5,17,18], we investigated whether the loss of PTEN, known to be involved in apoptotic pathways [19], might counteract the cell death in FGFR-deficient lens cells. These experiments showed that the loss of *Pten*, although not eliminating apoptosis completely, substantially rescued cell survival in *Fgfr2*-deficient lens cells [20]. The antagonistic relationship of FGFR-signaling and PTEN was also demonstrated in osteoprogenitor cells [21], and in the skin where the loss of *Fgfr2* could block skin tumor formation in *Pten*-deficient keratocytes [22,23]. 

To investigate the relationship of FGFR-signaling and PTEN during lens fiber cell differentiation, we adopted the use of the lens epithelial explant system. This system permits easy manipulation of culture conditions to study the regulation of fiber cell differentiation in vitro [1,24,25,26]. Using this system, we were able to employ adenoviral vectors to delete *Fgfrs* and/or *Pten* in the postnatal lens epithelium, thus avoiding their deletion at the earlier (embryonic) stages where FGFR-signaling is essential for lens epithelial cell survival. These experiments demonstrated that the loss of *Fgfrs* blocked the ability of vitreous humor to differentiate lens epithelial cells in vitro, as measured by cellular elongation, the accumulation of MIP and β- and γ-crystallins. However, the additional deletion of *Pten* in conjunction with *Fgfr*-loss restored the ability of vitreous humor to induce lens fiber cell differentiation [25]. Therefore, PTEN normally blocks the ability of non-*Fgfr* RTKs from inducing lens fiber cell differentiation.

We recently applied transcriptomic analysis to vitreous-induced lens fiber cell differentiation in wildtype lens epithelial explants from *FVB* strain mice [24]. Our analysis confirmed that the addition of vitreous humor resulted in the upregulation of many genes associated with lens fiber cell differentiation and a downregulation of many genes associated with lens epithelial cells. We also discovered that the addition of vitreous humor induces the expression of many genes associated with inflammation and epithelial to mesenchymal transition (EMT). However, the global changes underlying the deletion of *Fgfrs* and/or *Pten* in the vitreous-induced fiber differentiation in the lens epithelial explant system have never been addressed. Therefore, our purpose here is to provide a comprehensive transcriptomics view, rather than selective examination of a few differentiation markers, of vitreous humor-induced lens fiber cell differentiation and inflammation in the absence of *Fgfrs* or *Pten* or both *Fgfrs* and *Pten*. 

Our analysis demonstrates that the loss of FGFR-signaling profoundly inhibits the ability of vitreous humor to induce fiber cell differentiation. Further, it reduces the expression of genes involved in the inflammatory response. Consistent with our previous report, we found that vitreous exposure induced a better differentiation response in explants lacking both *Fgfrs* and *Pten* compared to explants lacking only *Fgfrs* [25]. Thus, concomitant loss of *Pten* can rescue lens fiber-differentiation defects resulting from loss of *Fgfrs*. Likewise, we found that the loss of *Pten* in *Fgfr*-deficient explants partially restored the vitreous-induced inflammatory response and resulted in an EMT response even higher than that seen in wildtype explants. Bioinformatic analysis of the gene-expression changes, induced by vitreous humor exposure, in explants lacking only *Fgfrs* or explants lacking both *Fgfrs* and *Pten* suggested the involvement of PDGFR-signaling in the restoration of differentiation in the explants deficient in both *Fgfrs* and *Pten*, thus identifying a mechanistic insight into the molecular basis of the rescue. 

## 2. Materials and Methods

### 2.1. Animals

The Miami University Institutional Animal Care and Use Committee reviewed and approved all described procedures. Animal use also complied with both the ARVO Statement for the Use of Animals in Research, and The Guide for the Care and Use of Laboratory Animals, published by the Institute for Laboratory Animal Research. *FVB/N* mice were euthanized by CO_2_ asphyxiation at postnatal day 8 (P8).

### 2.2. Lens Epithelial Explant Generation and Culture

Lens epithelial explants were established from 8-day-old (P8) mouse pups from four different genotypes. These were: wildtype *FVB/N* strain control (FVB), *Fgfr1^L/L^Fgfr2^L/L^Fgfr3^−/−^* (TKO), *Fgfr1^L/L^Fgfr2^L/L^Fgfr3^−/−^/Pten^L/L^* (QKO) and *Pten^L/L^* (PTEN), where the superscript “*L*” refers to a LoxP-flanked allele and superscript “−” refers to a null allele. Lens epithelial explants were established as previously described [24,25,26,27]. Briefly, lenses were removed from the eye and placed into a 35 mm culture dish with pre-calibrated M-199 media (37 °C and 5% CO_2_), supplemented with 1% antibiotic-antimycotic solution (ThermoFisher Scientific, Waltham, MA, USA, cat. 15240062) and 0.1% bovine serum albumin (Sigma-Aldrich, St. Louis, MO, USA, cat. A7906), henceforth referred to as “culture medium (CM)”. After tearing the lens capsule at the posterior pole, the fiber cells were removed. The lens capsule with adherent epithelium was then pinned to the culture dish. Immediately after explant establishment (6 explants per dish), recombinant *Cre*/GFP adenoviral vectors (Ad-GFP-2A-iCre, Vector BioLabs, Malvern, PA, cat. 1772 at 1 × 10^6^ PFU per dish) were added to the TKO, QKO and PTEN explants to mediate gene deletion. Adenoviral transduction was confirmed by observing GFP expression 24 h later.

Fresh bovine vitreous humor (provided by Kaiser Meat Market, Cedar Grove, IN, USA) was collected as previously described [28] and stored at −20 °C. One day (24 h) after the establishment of explants in CM, the medium was changed to vitreous humor, diluted 50% (*v*/*v*) with culture media, henceforth referred to as “differentiation medium” (DM). Following the addition of DM, explants were collected after 1, 5 or 10 days of culture.

### 2.3. RNA-Seq Library Preparation and Sequencing

RNA-seq analyses were performed for each condition with three biological replicates, each consisting of 12 explants. Total RNA was extracted from lens explants using the RNEasy Mini Kit (Qiagen, Hilden, Germany, cat. 74104), following the manufacturer’s instructions. RNA quality was assessed using an Agilent 2100 Bioanalyzer and samples with an RNA Integrity Number (RIN) >7 were used for sequencing. RNA libraries were constructed by Novogene (Sacramento, CA, USA) using a TruSeq (Illumina, San Diego, CA, USA) library prep kit. The library was sequenced on the Illumina NovaSeq sequencer platform. Sequences consisted of paired-end reads to a minimum depth of 26 million reads per sample.

### 2.4. RNA Seq Data Analysis

Raw read sequences were analyzed for quality using FastQC [29] and MultiQC [30]. Adapters and low-quality bases were trimmed using Cutadapt 3.4 [31] and Trim Galore 0.6.5-1 with the parameters -q 20 --phred33 --length 20 Babraham Bioinformatics-Trim Galore! Available online: https://www.bioinformatics.babraham.ac.uk/projects/trim_galore accessed on 20 February 2024. Mouse genome GRCm39 version: M27 was indexed using Hisat2 (2.1.0-4) [32], incorporating splice junctions from Gencode GTF gencode.vM27.annotation.gtf file [33]. Stringtie 2.1.5 [34] and gencode GTF annotation gencode.vM27.annotation.gtf was used to generate gene counts. DESeq2 1.34.0 [35] was used to determine differential expression. Differentially expressed genes (DEGs) were defined by an adj. *p*-value ≤ 0.005, and log_2_ fold change (LFC) ≥ 1.5. For DEGs, the *p*-value was adjusted using Benjamin-Hochberg correction individually for each pairwise comparison. The sequence data for *FVB* explants at D1 and D5 were taken from GEO Series accession number GSE215953. 

### 2.5. Clustering of DEGs

K-means clustering (part of the “stats” package in R), using normalized RNA-seq counts [36], was used to cluster DEGs. The kmeans function was used with the Hartigan–Wong algorithm [37]. Metascape [38], was used for pathway enrichment. Some of the Gene Ontology figures are directly adapted from metascape. Individual K-means clusters were used for GO term analysis. All figures were generated in the R 4.1.3 environment. Venn diagrams were made using Venny [39].

### 2.6. Gene Set Enrichment Analysis (GSEA) and Protein Protein Interaction

The normalized count matrix obtained from DESeq2 was used for Gene set enrichment analysis (GSEA v4.3.2), and murine genes were converted to human orthologs [40,41]. GSEA was performed using 1000 permutations and gene set permutations with gene set size filters: min = 15 and max = 500. Both the Hallmark gene set and Reactome C2 gene set were used for analysis. 

The online STRING database (version 12.0) was used for generation of protein interaction for the selected gene sets [42]. Following parameters were used for visualization network type = Full STRING network, Meaning of network edge = Confidence, Active interaction source = All selected, Minimum required interaction score = medium confidence (0.400).

### 2.7. Lens Explant Immunofluorescence

Lens explants were fixed in 4% paraformaldehyde for 5 min and then washed three times in PBS followed by three washes in 0.1% PBST (each for 5 min). Cytoskeletal actin was then visualized following incubation with Alexa Fluor 488 conjugated phalloidin (Thermo Fisher Scientific Waltham, MA, USA, cat. 12379) for 20 min at room temperature. β-crystallin and MIP was revealed by fixing explants in methanol for 1 min before PBS washes and blocking with 10% normal goat serum in PBS at 37 °C for 30 min. Primary antibodies included: rabbit polyclonal anti β-crystallin (gifts from Samuel Zigler, Johns Hopkins University School of Medicine, Baltimore, MD, USA) and rabbit polyclonal anti-Aquaporin 0 (Abcam Cambridge Guildhall, UK, cat. ab15077) for MIP. Anti β-crystallin antibody and anti-Aquaporin 0 was used at a 1:100 dilution and incubated at 4 °C overnight in 10% normal goat serum in 0.1% PBST. Fluorescent secondary antibodies (goat anti-mouse Alexa Fluor 594 or goat anti-rabbit Alexa Fluor 488, ThermoFisher Scientific, Waltham, MA, USA, cat. A21125 and A1108) detected primary antibody binding. Following counterstaining with DAPI (Vector Labs, Malvern, PA, USA, cat H-1200), fluorescent images were captured using the Zeiss 710 Laser Scanning Confocal System at the Center for Advanced Microscopy and Imaging at Miami University. 

For inhibitor treatment, explants were pre-treated for two hours with either 25 μM of the selective PI3K inhibitor LY294002 (Selleckchem, Houston, TX, USA, cat. S1105), 50 μM of the selective MEK1/2 inhibitor U0126 (Selleckchem, cat. S1102) or 20 μM AG1296 of the selective PDGFR inhibitor and 5 μm AG1024 (Selleckchem, cat. S1234) of IGF-1 inhibitor and remained present with the differentiation media for the duration of the culture period.

### 2.8. RT-qPCR

To validate RNA-Seq and immunofluorescence results, selected genes were analyzed using RT-qPCR. RNA extraction followed the previously described protocol, and RNA was reverse transcribed into cDNA with oligo (dT) and random primers, following the ImProm-IITM Reverse Transcription System, Madison, WI, USA, cat. A3800 manufacturer’s instructions. RT-qPCR assays were conducted with three biological replicates and three technical replicates using GoTaq qPCR Master Mix, Madison, WI, USA, cat. A6001 and the Bio-Rad CFX Connect. Pre-designed primers were sourced from PrimerBank (https://pga.mgh.harvard.edu/primerbank/, accessed on 16 May 2024) and obtained from IDT Technologies, Coralville, IA, USA (Appendix A)

The comparative ∆∆Ct method was used to determine relative gene-expression levels compared to housekeeping genes (*Gapdh*) and relative mRNA was normalized to FVB. To determine the statistical significance of the observed differences in gene expression, we performed an analysis of variance (ANOVA) on the logFC values, considering the effects of treatment and gene. This was followed by Tukey’s Honest Significant Difference (HSD) post-hoc test to identify specific group differences. The results of the Tukey HSD test were used to annotate the plots with significance levels, indicated as ‘ns’ for non-significant, * for *p* < 0.05, ** for *p* < 0.01 and *** for *p* < 0.001.

## 3. Results

### 3.1. Overall Transcriptomic Changes in Lens Epithelial Explants Correlated Both to Genotype and Duration of Exposure to Vitreous Humor

A comprehensive RNA-seq analysis was undertaken on lens epithelial explants from multiple genotypes: wildtype (FVB), those lacking *Fgfrs* (TKO), those lacking *Pten* (PTEN) and those lacking both *Fgfrs* and *Pten* (QKO) at several different time points. These time points were (A) 24 h after the addition of DM, containing vitreous humor (D1), (B) 5 days after the addition of DM (D5) and 10 days after the addition of DM (D10) (Figure 1A). The D1 and D5 time points correspond to the induction of differentiation and the onset of differentiation gene expression (corresponding to our previous paper describing vitreous-induced differentiation in *FVB/N* explants [24]) and D10 corresponds to late differentiation phase. The resultant data, consisting of approximately 26 million sequence reads per biological replicate, was represented by three-dimensional principal component analysis (3D-PCA) (Figure 1B). The first three components PC1, PC2 and PC3 were plotted as X, Y and Z axes to observe the major components shaping each dataset with three replicates per genotype and treatment. The replicates within each genotype and treatment consistently clustered together underscoring a high concordance between biological replicates. The overall similarity of the biological replicates and the effect of treatments was also visualized by a distance matrix heatmap of the RNA-seq data (Appendix A). The relative closeness of gene expression among the replicates within various treatment groups suggests that data were overall of robust quality.

### 3.2. Loss of FGFRs, PTEN or Both FGFRs and PTEN Induced Different Patterns of Differential Gene Expression Following Vitreous Exposure

To visualize patterns of differential gene expression within each genotype based on different stages in fiber differentiation (as determined by the time since initiation of incubation in DM), we performed K-means clustering using a series of pairwise comparisons: Day 1 vs. Day 5, Day 5 vs. Day 10 and Day 1 vs. Day 10 for TKO, QKO and PTEN, respectively. All non-redundant differentially expressed genes (DEGs) with log_2_ fold change ≥ 1.5 and *p*-adjust value ≤ 0.005 were analyzed using this approach to distinguish five patterns of gene expression within each genotype (Figure 2).

The K-means clustering of DEGs for explants lacking Fgfrs (TKO) revealed 5 clusters for a total of 1639 DEGs (Figure 2A). Selected gene ontology (GO) terms for each cluster are shown in a bubble plot (Figure 2D) with a full list of GO terms associated with each cluster (T1–T5) in Appendix A. On day 1, the genes in cluster T1 exhibited their peak level of expression. T1 consists of 223 genes that are enriched for GO terms including “Epithelial cell differentiation” (exemplified by *Atf4*, *Bmp2*, *Cdh1*). The genes in cluster T3 exhibited their lowest level of expression on day 1. T3 consists of 388 genes that are enriched for the GO terms “Extracellular matrix organization” (exemplified by *Bfsp1*, *Col12a1*, *Mmp12*), “Regulation of Phosphatidylinositol 3-kinase/protein Kinase B signal transduction” (exemplified by *Gas6*, *Ptk2b*, *Serpine2*) and “Positive regulation of MAPK cascade” (exemplified by *Igf1*, *Pdgfb*, *Pdgfc*). On day 5, the genes represented by cluster T5 reached their expression peak. T5 consists of 608 genes that are enriched for GO terms including “Regulation of Inflammatory response” (exemplified by *Alox5*, *C3*, *Csfr1*) and “Positive regulation of immune response” (exemplified by *Bkt*, *Fcgr3*, *Tlr7*). The genes in cluster T4 exhibited their lowest expression level on day 5. T4 consists of 200 genes that are enriched for GO terms including “Apoptotic-signaling pathway” (exemplified by *Bdkrb2*, *Cdkn1a*, *Trib3*). In contrast, genes in clusters T3 and T5 reached their expression peak on day 10 while genes in cluster T2 fell to their lowest expression level on day 10. T2 consists of 220 genes that are enriched for GO terms including “Protein digestion and Absorption” (exemplified by *Egfl6*, *Fxyd2*, *Prss1*), “Monoatomic ion transmembrane transport” (exemplified by *Abcc3*, *Cacna1e*, *Muc5ac*). This suggests that in the absence of FGFR-signaling, the lens epithelial explants reach a peak inflammatory response at D5 and undergo peak PI3-kinase and MAPK-signaling responses at D10. However, we do not see the induction of gene-expression changes specific for lens fiber cell differentiation. 

Similarly, clustering of DEGs for QKO revealed five clusters of DEGs for a total of 1708 genes (Figure 2B). Selected GO terms for different clusters are shown in Figure 2D with a full list of GO terms associated with each cluster (Q1–Q5) in Appendix A. On day 1, the genes in cluster Q2 reached their highest level of expression. Cluster Q2 consists of 485 genes enriched for GO terms including “Mitotic cell cycle” (exemplified by *Bard1*, *Cdk1*, *Mcm2*), and “Positive regulation of cell cycle” (exemplified by *Aurka*, *Bric5*, *E2f7*). The genes in clusters Q1 and Q5 exhibited their lowest level of expression on day 1. Cluster Q1 consists of 417 genes enriched for GO terms including “Lens development in camera-type eye” (exemplified by *Cryba1*, *Gja3*, *Mip*), “Lens fiber cell differentiation” (exemplified by *Bfsp1*, *Tdrd7*, *Tmod1*) and “Positive regulation of MAPK cascade” (exemplified by *Adra2a*, *Cd84*, *Fzd10).* Cluster Q5 consists of 433 genes enriched in GO terms including “Activation of immune response” (exemplified by *Casp1*, *C3ar1*, *Mef2c*), “Degradation of extracellular matrix” (exemplified by *Bcan*, *Col2a1*, *Eln*) and “Neutrophil degranulation” (exemplified by *Arg1*, *Cd33*, *Dock2*). On day 5, genes in cluster Q5 reached their peak level of expression while genes in cluster Q3 exhibited their lowest expression level. Cluster Q3 consists of only 98 genes with no interesting GO term. Genes in clusters Q1 and Q3 exhibited their highest level of expression on day 10, while genes in clusters Q4 and Q2 exhibited their lowest level of expression on that day. Cluster Q4 consists of 275 genes enriched for GO terms including “Transmembrane receptor protein tyrosine-signaling pathway” (exemplified by *Efs*, *Flt4*, *Ret*), “Retina development in camera-type eye” (exemplified by *Neurod4*, *Otx2*, *Vsx2*) and “Epithelial cell differentiation” (exemplified by *Dll1*, *Krt14*, *Neurod1*). Overall, the QKO explants exhibit responses to DM similar to wildtype explants, including an initial burst of proliferation (D1) and maximal expression of genes related to fiber cell differentiation on D10.

Clustering of DEGs for PTEN revealed five clusters, encompassing a total of 3358 genes (Figure 2C). Representative GO terms for different clusters are shown in Figure 2D with a full list of GO terms associated with each cluster (P1–P5) in Appendix A. On day 1, the genes in cluster P3 reached their peak expression level. Cluster P3 consists of 470 genes enriched in GO terms including “Negative regulation of cell differentiation” (exemplified by *Btg2*, *Cdh1*, *Map2*), “Regulation of nervous system development” (exemplified by *Dkk1*, *Map2*, *Mme*) and “Regulation of Wnt-signaling pathway” (exemplified by *Folr1*, *Nog*, *Wnt6*). In contrast, genes in clusters P1, P2 and P4 exhibited their lowest level of expression on day 1. Cluster P1 contains 567 genes that are associated with GO terms including” Lens development in camera-type eyes” (exemplified by *Crybb3*, *Crygs*, *Lim2*), and “PRC2 methylates histone and DNA” (exemplified by *H2bc4*, *Shc2*, *Timeless*). Cluster P2 contains 494 genes and is enriched for GO terms including “Extracellular matrix organization” (exemplified by *Col11a1*, *Mmp10*, *Nr2e1*), “Positive regulation of cell migration” (exemplified by *Cxcr4*, *Nrp1*, *Pdgfd*) and “Regulation of immune effector processes” (exemplified by *Btk*, *Casp1*, *Cd84*). Cluster P4 consists of 1284 genes that are enriched for GO terms including “Inflammatory response” (exemplified by *Adora2*, *Ccr1*, *Fcgr3*) and “Regulation of MAPK cascade” (exemplified by *Cav2*, *Fn1*, *Fzd7*). On day 5, genes in clusters P2, P4 and P5 reached their peak expression level. Cluster P5 consists of 543 genes that are associated with GO terms including “cell junction organization” (exemplified by *Cacnb2*, *Cdh5*, *Wnt11*), “Regulation of BMP-signaling pathway” (exemplified by *Bmp4*, *Cav1*, *Shisa2) and* “Sensory organ development” (exemplified by *Crx*, *Rax*, *Otx1*). While genes in cluster P5 peaked at day 5, these genes reached their lowest level of expression on day 10. Day 10 also experienced a peak of gene expression in cluster P1. Overall, in the PTEN explants, there is an initial suppression of MAPK signaling and inhibition of differentiation at D1, followed by a peak of extracellular matrix organization, cell migration and immune response at D5 and a strong fiber cell-differentiation response at D10. 

### 3.3. FGFRs and Pten Influence Gene Expression in Opposite Directions with Respect to Fiber Cell Differentiation and Chromatin Remodeling Following DM Exposure

The lens epithelial explants system was developed to study lens fiber cell differentiation in vitro. To evaluate how the loss of FGFRs affect vitreous-induced lens fiber cell differentiation, we compared the expression of 25 genes (based on genes differentially expressed at a higher level in lens fiber cells than in lens epithelial cells [43,44]) characteristic of fiber cells in explants of each genotype over the 10-day culture period (Figure 3A). These genes encoded: nine crystallins, two gap junction proteins, major lens membrane proteins (MIP and LIM2), the transcription factor HSF4 and proteins involved in organelle degradation (DNASE2B and BNIP3) and other candidates. Nearly all of the 25 genes exhibit a progressive increase in gene expression in the control FVB explants following the addition of the vitreous-containing DM media. This was true for all of the crystallins, *Hsf4*, *Mip*, *Dnase2b*, *Lim2* and *Bnip3*. The loss of *Fgfrs* in the TKO samples significantly reduced the expression of these genes. Notably, there was a dramatic loss of expression for *Gja8*, *Gja3*, *Nav3*, *Crybb3*, *Lgsn*, *Jag1* and *Pla2g7* 10 days after DM addition. The expression of nearly all of these genes increased in the QKO samples relative to the TKO samples. In fact, the additional loss of *Pten* (in the QKO samples relative to the TKO samples) led to a level of expression of these fiber cell genes in DM-induced samples at day 10 to approximately the level seen in the control FVB explants on day 5. 

This analysis reinforces the notion that PTEN counterbalances FGFR signaling during lens fiber cell differentiation. These data show the loss of PTEN in explants lacking FGFRs is sufficient to rescue aspects of gene expression associated with a fiber cell-differentiation response, which only increases with time in DM culture. Consistent with this idea, the loss of *Pten* alone led to an enhanced differentiation response in explants following DM exposure (as measured by fiber gene expression). This is particularly notable at day 1 where all other genotypes exhibited a relatively low level of expression for the majority of the fiber cell genes. An exception to this pattern were the levels of expression of *Pla2g7*, *Slc2a3*, *Jag1* and *Lgsn* that were expressed at low levels at day one but increased during days 5 and 10. In all genotypes, with the exception of the TKO, the expression of the majority of the fiber cell genes was most robust at day 10 following DM exposure, suggesting that the differentiation response in culture is progressive. 

The response to DM of lens epithelial explants, with respect to the expression of genes characteristic of lens epithelial cells [24], was also analyzed across the different genotypes (Figure 3B). These genes were divided into three groups, group I (4 genes), group II (6 genes) and group III (10 genes), based on differential responses to DM. The genes in group I (*Mki67*, *Cdk1*, *Notch4* and *Flt1*) were notable for their low expression at day 1 in the *Pten* knockout explants. The expression of all these genes, with the exception of *Notch4*, increased subsequent to this timepoint in the PTEN samples. The genes in group II (*Kdr*, *Dll4*, *Rgs6*, *Dll1*, *Sulf1* and *Gja1*) were generally expressed at a higher level in the TKO explants at all timepoints (consistent with these explants being refractory to DM-induced differentiation) and the expression of these genes fell dramatically in the PTEN explants at day 10 (consistent with a very strong DM-induced differentiation response). The genes in group III (e.g., *Foxe3*, *Cdh1*, etc.) were expressed at a low level in the FVB explants irrespective of time, suggesting the DM exposure induced the wildtype explants to lose epithelial gene expression from initial stages. In contrast, group III genes exhibited the highest expression level at day 1 in the PTEN explants. The PTEN explants exhibited a subsequent sharp reduction in the expression of the group III genes, with the exception of *Foxe3* and *Pdgfra* where the expression remained moderately high on days 5 and 10. Group III genes maintain a moderate expression level in both the TKO and QKO explants throughout DM exposure. Interestingly, the genes *Mme*, *Foxe3* and *Pdgfra* exhibit a dramatic expression decline at day 10 in TKO explants. Overall, the absence of *Fgfrs* helped epithelial cells to maintain the expression of genes consistent with lens epithelial identity. However, the loss of *Foxe3* and *Pdgfra* might also suggest a progressive loss of lens cell identity in DM when *Fgfrs* are deleted.

Recently, a single nucleus (sn)-multiomics analysis of the developing chick lens suggested that the PRC2 complex may play an important role in lens fiber cell differentiation [45]. Gene ontology analysis on cluster 1 (P1) from the PTEN explants suggested a possible link between PTEN and genes associated with PRC2. Therefore, we surveyed all 79 of the genes identified by the gene ontology program Metascape using the GO term “PRC2 methylates histones and DNA”. Half of these (38 genes) were differentially expressed in the PTEN explants, and the expression of these genes was analyzed across all explant genotypes (FVB, TKO, QKO and PTEN) for all timepoints (Appendix A). In general, the expression of these genes was low across genotypes on day 1, with the expression being slightly more elevated in the TKO explants and markedly more depressed in the PTEN explants. Exceptions to this pattern were (*Timeless*, *Gtse1*, *Tubb4b*, *H2ac12* and *Pidd1*) where the expression on day 1 was high in FVB, and QKO genotypes but low in the TKO genotype and very low in the PTEN genotype on day 1. Most of these genes increased in expression with DM exposure, peaking on day 10 in the FVB, QKO, and PTEN genotypes. However, most of these genes remained relatively unresponsive to DM exposure in the TKO genotype, except for *Ddit4* and *Usp2*, which increased with DM exposure in the TKO samples.. Notably, the group of genes previously highlighted (*Timeless*, *Gtse1*, *Tubb4b*, *H2ac12* and *Pidd1*) decreased in expression from day 1 to day 10 in the FVB and QKO genotypes while the expression of these genes increased slightly in the TKO genotype and increased markedly in the PTEN genotype through this time period. Overall, the most impressive feature among these genes was the dramatic increase in the expression of these genes from day 1 to day 10 in the PTEN explants. Further, removal of *Pten* (in the QKO) is sufficient to restore the expression of many of these genes to levels similar (or higher) to that in the FVB (control) explants. These observations are consistent with the notion that FGFR signaling increases and PTEN suppresses the expression of these genes in response to DM. Since peak differentiation-related gene-expression responses occurred at day 10 following DM exposure, we focused on this period for further analysis.

### 3.4. Epistasis between FGFR-Signaling and PTEN in Lens Epithelial Explants in Response to DM

Our previous studies had demonstrated that (1) the loss of *Pten* could counter the apoptosis induced by the loss of *Fgfr2* in the lens [20] and (2) the loss of *Pten* could restore the ability of vitreous to induce fiber cell differentiation in *Fgfr*-deficient lens epithelial explants [25]. These observations suggest that normally FGFR-signaling promotes, and in the absence of *Fgfrs*, PTEN inhibits lens fiber cell differentiation, consistent with the data shown in Figure 3A. Since the greatest fiber cell-differentiation response in the FVB, QKO and PTEN explants was seen 10 days after DM treatment, we chose to focus on this time point for further analyses. To determine the overall epistatic relationship of *Fgfrs* and *Pten* with respect to DM, we analyzed how the expression of genes in the QKO genotype is modified relative to that in the TKO and PTEN genotypes at day 10. Relative to the wildtype FVB genotype at day 10, the TKO explants deregulate 1343 genes, the QKO explants deregulate 1687 genes and the PTEN explants deregulate 2645 genes. To illustrate how *Fgfrs* and *Pten* interact genetically at day 10, we constructed Venn diagrams to demonstrate the restoration of normal gene expression in both the TKO and PTEN genotypes in the QKO genotype (Figure 4). Of the 683 genes downregulated in the TKO genotype, 479 (70%) of these were restored to normal regulation in the QKO genotype. Of the TKO downregulated genes that remained deregulated in the QKO genotype, 198 of these remained downregulated and 6 were upregulated. Likewise, of the 660 upregulated genes in the TKO genotype, 518 (78%) were regulated normally in the QKO genotype. Of the 142 TKO upregulated genes that remained deregulated in the QKO explants, 125 of these remained upregulated and 17 were downregulated (Figure 4A). Of the 2359 genes that were downregulated at day 10 in the PTEN genotype, 1605 (68%) of these were normalized in the QKO genotype. Of the PTEN downregulated genes that remained deregulated, 739 of these remained downregulated and 15 were upregulated in the QKO genotype. At day 10, the PTEN explants upregulated 286 genes. Of these, 165 (58%) were rescued to normal regulation in the QKO genotype. Of the 121 genes that remained deregulated in the QKO genotype, 113 remained downregulated and 8 were upregulated (Figure 4B). It is interesting to note that while the loss of *Fgfrs* resulted in almost an equal number of upregulated (49%) and downregulated (51%) genes at day 10, both the PTEN and QKO genotypes downregulated (89% and 80%, respectively) more genes than they upregulated (11% and 20%, respectively). All of the relevant genes described in Figure 4 are included in Appendix A. The simultaneous deletion of *Pten* and *Fgfrs* in the QKO led to the restoration of normal expression in 70% and 78% of the genes downregulated or upregulated, respectively, in the explants lacking only *Fgfrs* (TKO). 

Since we were most interested in the moderation of the TKO gene-expression changes by the additional loss of *Pten*, we normalized the PTEN and QKO genotypes with respect to the TKO genotype at day 10 and visualized deregulated gene expression with a Venn diagram (Appendix A). Most interesting were the intersections of the 862 genes that were upregulated in both the PTEN and QKO explants and the 1107 genes that were downregulated in both the PTEN and QKO explants at day 10. GO analysis of the commonly upregulated genes (Appendix A) revealed the term “lens development in camera-type eye” as most significant (padj = 3.95 × 10^−15^). Other notable terms from this category included “actin filament-containing process” (padj = 3.67 × 10^−10^), “regulation of epithelial cell proliferation” (padj = 3.13 × 10^−8^), “signaling by GPCR” (padj = 8.86 × 10^−8^) and “extracellular matrix organization” (padj = 1.68 × 10^−7^). The GO analysis of the 1,107 commonly downregulated genes (Appendix A) revealed the terms “regulation of tube size” (padj = 8.31 × 10^−3^), “neuronal system” (padj = 1.69 × 10^−2^). These data also suggest that the commonly downregulated genes include those functioning in neurons, which is in agreement with reports suggesting that there are mechanisms that are at play for downregulating neuronal gene expression in lens fiber cells [45,46]. Here it appears that the 862 commonly upregulated genes in the QKO and PTEN genotypes are those most likely to participate in lens fiber cell differentiation. Of these commonly upregulated genes, 234 overlap with the 479 genes downregulated in the TKO genotype that are restored to normalcy in the QKO genotype. The top GO term for these 234 genes was “lens development in the camera-type eye” (padj = 1.96 × 10^−13^) and the second highest GO term was “lens fiber cell differentiation” (padj = 3.02 × 10^−6^). It is likely that the genes most responsible for the phenotype in the TKO explants are contained in this list of 234 genes (Appendix A, Appendix A). 

### 3.5. The Effect of Fgfr/Pten Epistasis on the Expression of Genes Listed in iSyTE and Cat-Map

To gain a greater understanding of the effect of *Fgfr* and/or *Pten* loss on all aspects of lens development, we analyzed our day 10 transcriptomic data in light of the 528 lens-enriched genes listed in the iSyTE dabase [47] and the list of 496 cataract-associated genes maintained by the Cat-Map database [48]. A total of 69 genes are shared between the iSyTE and Cat-Map lists. Among the 1343 deregulated genes in the TKO genotype (relative to the wildtype FVB genotype), 123 (9%) were shared with either or both the Cat-Map and iSyTE database (Figure 5A). In contrast, only 89 (3%) of the 2645 deregulated genes from the PTEN genotype were shared with either or both of these database lists (Figure 5B). The QKO genotype deregulated 46 (3% of the total 1687 deregulated genes) genes that were on either or both of the Cat-Map or iSyTE gene lists at day 10 (Figure 5C). If we combine the iSyTE and Cat-Map genes into a group that will hereafter be designated “Lens Genes”, five of these lens genes remain deregulated in all three genotypes (TKO, QKO and PTEN) relative to FVB. These are *Mfrp*, *Car8*, *Cdh1*, *Myb* and *Cckbr*. With respect to transcription factor genes, *Foxe3* remains deregulated whenever Pten is deleted (PTEN and QKO) and *Otx2* is deregulated whenever the Fgfrs are deleted (TKO and QKO). The only transcription factor gene that is commonly deregulated in the TKO and PTEN genotypes is *Sox1*, and the expression of this gene normalizes in the QKO genotype. Three genes (*Aldh1a7*, *Sox1* and *Gabrg3*) were both downregulated in the TKO genotype and upregulated in the PTEN genotype). Likewise, four genes (*Cckbr*, *Alpl*, *Sned1* and *Slc4a50* are both upregulated in the TKO and downregulated in the PTEN genotypes. Interestingly, no iSyTE genes were commonly downregulated in the TKO genotype and upregulated in the QKO genotype or upregulated in the TKO genotype and downregulated in the QKO genotype. These data suggest that QKO showed the restoration of the majority of the iSyTE/Cat-Map genes found to be deregulated in TKO.

### 3.6. PDGFRa: A Potential Alternative Pathway for Fiber Cell Differentiation in Absence of FGFRs and PTEN

We previously showed that the removal of *Pten* can restore the ability of vitreous humor to elicit lens epithelial cell elongation and the expression of both β-crystallin and MIP in lens epithelial cells lacking *Fgfrs* [25]. Our current gene-expression analysis also supports the ability of DM to induce gene-expression changes—including the restoration of β-crystallin and *Mip* genes, among others—consistent with fiber cell differentiation in the absence of *Fgfrs* when *Pten* is deleted. To determine if our transcriptomics data might provide evidence as to the mechanism by which *Fgfr*-independent differentiation occurs, we employed gene set enrichment analysis (GSEA) to conduct a comprehensive pairwise comparison to predict possible pathways driving fiber cell differentiation in absence of both *Fgfrs* and *Pten.* Gene Set Enrichment Analysis (GSEA) is a computational method used in bioinformatics to determine whether predefined sets of genes exhibit statistically significant differences in expression between two biological states [40,41]. Since gene expression suggesting fiber cell differentiation was most prominent in D10 explants, we compared the TKO and QKO gene-expression data at this stage using the GSEA C2 reactome dataset to identify differential enrichment for well-defined biological pathways. Among the highest ranking pathways enriched in the QKO explants relative to the TKO explants were those involved in the cholesterol and extracellular matrix, such as “Anchoring fibril formation”, “Cross linking of collagen fibrils”, “Cholesterol biosynthesis”, “Laminin interactions”, “Assembly of collagen fibrils and other multimeric structures”, “Metabolism of steroid hormones”, “Regulation of cholesterol biosynthesis and activation of gene expression by SREBP/SREBF” and “Non-integrin membrane ECM interactions” (Appendix A). However, an interesting pathway that appeared in the QKO and TKO GSEA analysis was “signaling by PDGF” with an enrichment score of 0.5357 (Figure 6A) based on the enrichment of 27 genes. This was the only enriched term that pointed to a receptor tyrosine kinase pathway related to FGFR-signaling. This led to the speculation that in the absence of *Pten*, DM-induced PDGF receptor (PDGFR)-signaling might rescue fiber cell differentiation in *Fgfr*-deficient lens epithelial explants.

To investigate the possibility that PDGFR-signaling contributes toward rescuing aspects of fiber cell differentiation in *Fgfr*-deficient explants, we analyzed the genes enriched in QKO, relative to TKO explants, with respect to all the genes listed under the reactome pathway term “signaling by PDGF”, and with respect to differential expression in all other genotypes at all time points. Of the 58 total genes in the “signaling by PDGF” pathway, 27 (~47%) were enriched in the QKO explants relative to the TKO explants at D10. Of these 27 genes, 17 (~63%) were differentially expressed (log_2_fold > 1.5, padjust < 0.005), relative to the wildtype explants, in at least one genotype or time point following DM exposure (Figure 6B). By definition, all 27 of these genes were expressed at a higher level in the QKO explants than the TKO explants at D10 but not all of these genes had to meet the criteria we set for DEGs. For example, a QKO enriched gene in GSEA analysis might not have met the log_2_ > 1.5, padjust < 0.005 threshold for DEG. The 17 genes that did meet the DEG criteria were further divided into three categories (I, II, III) based on similar gene-expression patterns in the different genotypes. 

The first category (I) contains eight genes (*Pdgfra*, *Stat5a*, *Col4a2*, *Col4a1*, *Col4a3*, *Col4a4*, *Stat1* and *Col6a3*). All of the genes in this category, with the exception of *Pdgfra*, increase from D1 to D10 in the wildtype FVB explants. Within the D10 samples, all of these genes are expressed at the lowest levels in the TKO explants. Also, at D10 all of these genes, with the exception of *Col4a3* and *Col6a3* are expressed at the highest level in the PTEN explants, with the two exceptions being expressed higher in the QKO explants. At D1, all of these genes are expressed at a higher level in the PTEN explants than in any other genotype. 

The second category (II) contains four genes (*Col6a1*, *Thbs4*, *Thbs3* and *Col9a3*). At D1, all of these genes are expressed at the lowest level in the PTEN explants and at the highest level in the TKO explants, with the exception of *Thbs4* which is expressed at the highest level in the QKO explants at D1. At D10 the expression level of these genes is similar in the FVB and TKO explants, with relatively higher expression in the QKO and lower expression in the PTEN explants. 

The third category (III) includes five genes (*Spp1*, *Pdgfc*, *Pdgfb*, *Col5a3* and *Plat*). These genes are expressed relatively lower in the PTEN explants at D1. Otherwise, the pattern of expression through time for these genes is more similar in the FVB and QKO samples and relatively lower in the PTEN and TKO samples. It is interesting to note that two of these genes in this category (*Pdgfc* and *Pdgfb*) are PDGF ligands.

Given that D10 is when the vitreous-induced differentiation in lens epithelial explants is most pronounced in the FVB (wildtype) condition, we examined the expression of these PDGF-related genes in the TKO and QKO explants at this stage (indicated by the dashed purple boxes in Figure 6B. All of these genes are expressed more highly in the QKO explants than the TKO explants at D10. The genes in categories I and II are expressed at a higher level in the QKO samples than in the FVB samples at D10 while the genes in category III are expressed at a similar or higher level in the FVB explants than the QKO explants. PDGFRa, presumably upstream of these PDGF-related genes, normally exhibits elevated expression in lens epithelial cells and relatively lower expression in fiber cells. Consistent with this, the expression of *Pdgfra* is lower in the FVB explants at D10 (when the differentiation response to vitreous is strongest) than it is at D5. Interestingly, despite the overall lack of differentiation response in the TKO explants to vitreous, the level of *Pdgfra* expression is even lower at D10 in the TKO explants than in the FVB explants, and this expression decline at D10 is largely prevented in the absence of PTEN (both in the QKO and PTEN explants). 

To illustrate the relationship of these 17 genes, we utilized STRING, a web-based database tool that helps identify known or predicted protein–protein interactions based on a provided gene list (Figure 6C). The thickness of the lines in the network diagram represents the confidence in interaction between proteins. PDGFRa occupies a central position in the network, with strong interactions with the ligands PDGFB and PDGFC as well as signaling molecules, STAT1 and STAT5a and the SPP1 transcription factor. The extracellular matrix proteins, (THBS3, THBS4 and all of the collagens) form an interacting network downstream of PDGFRa. To illustrate the connections of PDGFR-signaling, FGFR-signaling and PTEN with lens fiber cell differentiation, we used STRING to construct a protein–protein interaction network with the *Fgfr* genes, the *Pdgfr* genes, genes involved in ERK and AKT-signaling pathway and all the genes associated with lens epithelium (Appendix A) or lens fibers (Figure 3A) that are differentially expressed in our dataset (Appendix A). Although *Pax6* was not differentially expressed in our dataset, given its known connection to FGFR-signaling [49,50], and its central place in establishing lens cell fate [51,52], we included PAX6 as well. As illustrated in the figure, FGFR-signaling is more strongly connected to both ERK (Mapk1 and Mapk3) and Akt than is PDGFR-signaling, but both are strongly connected to PTEN. The connections tying both FGFR- and PDGFR-signaling to the genes in the lens node are through PAX6 and GJA1. Indeed, *Pax6* conditional knockout (*Pax6*^cKO^) exhibits significant reduction of *Pdgfra* transcripts in the lens tissue at E9.5 and E10.5 (Table 1). Further, mouse conditional knockout models for E2F1/E2F2/E2F3 (three gene deletion) or Notch2, which exhibit lens defects, also show significant reduction of *Pdgfra* transcripts in the lens at later embryonic or newborn stages (Table 1). These data suggest that lens defects are associated with reduced expression of *Pdgfra*. These connections provide a possible explanation for why PDGFR-signaling could facilitate fiber cell differentiation in lens epithelial explants when removed from repression by PTEN.

To functionally test whether inhibition of PDGF signaling could block the vitreous-based differentiation response in QKO explants, we set up an experiment to determine if β-crystallin and MIP expression (read-out markers for differentiation) are impacted by the PDGFR inhibitor, AG1296. As we have shown before, the DM media, containing 50% bovine vitreous humor, elicits β-crystallin expression after 5 days in both FVB and QKO explants (Figure 7 and Appendix A). The addition of 20 μM AG1296 consistently (N = 15 explants) blocked the expression of β-crystallin and MIP in QKO explants. This effect was not generalized for the inhibition of non-FGFR receptor tyrosine kinases, because the inhibition of the IGF receptor (IGF-1R) with AG1024 failed to block the induction of these genes in QKO explants. Consistent with these observations, RT-qPCR analysis of *Crybb2* and *Crybb3* expression demonstrated that AG1296, but not AG1024, blocked the expression of these genes in the QKO genotype after 5 days of vitreous exposure (Appendix A). These findings provide a novel mechanistic basis (via PDGFR) of how *Pten* deletion results in the rescue of the fiber-differentiation defects resulting from the absence of *Fgfrs*.

### 3.7. ERK Signaling Is Essential for Fiber Cell Differentiation in Absence of FGFRs and PTEN

Since the removal of PTEN lifts the inhibition of vitreous-induced fiber cell differentiation in FGFR-deficient lens epithelial explants, we sought to determine the dependence of this rescue on ERK-signaling and/or AKT-signaling. To isolate the effect of activated AKT in the differentiation response, we treated explants with 25 μM of the selective PI3K inhibitor LY294002 for two hours prior to vitreous treatment in explants (Figure 8). LY294002 blocked the induction of β-crystallin accumulation in wildtype explants but did not affect β-crystallin accumulation in PTEN or QKO explants. We also measured *Crybb2* and *Crybb3* gene expression in FVB. TKO, QKO and PTEN explants in the presence of LY294002 by RT-qPCR (Appendix A). Consistent with the results in Figure 8, PI3K inhibition blocked *Crybb2* and *Crybb3* transcript expression in the FVB and TKO explants but failed to do so in the QKO and PTEN explants. In contrast, lens epithelial cell elongation, measured by phalloidin staining, was not blocked in wildtype, QKO or PTEN explants in the presence of LY294002 (Figure 8).

Likewise, to isolate the effect of activated ERK1/2 on fiber cell differentiation, we treated explants with 50 μM of the selective MEK1/2 inhibitor U0126 for two hours prior to vitreous treatment (Figure 9). MEK inhibition failed to block β-crystallin accumulation in wildtype or PTEN explants but did block β-crystallin accumulation in QKO explants. MEK inhibition prevented the vitreous-induced elongation in wildtype, PTEN and QKO explants. TKO explants do not accumulate β-crystallin or rearrange their cytoskeleton under any treatment. We also measured *Crybb2* and *Crybb3* gene expression in FVB. TKO, QKO and PTEN explants in the presence of U0126 by RT-qPCR (Appendix A). Consistent with the results in Figure 9, Mek inhibition blocked *Crybb2* and *Crybb3* transcript expression in the QKO and TKO explants but failed to do so in the FVB and PTEN explants. Therefore, neither β-crystallin accumulation nor elongation occurred in the QKO explants following MEK inhibition. Together, these findings show that signaling via ERK, rather than AKT, is necessary for the *Pten* deletion-based rescue of fiber-differentiation defects in the absence of *Fgfrs*.

### 3.8. The Removal of Pten Increased the Expression of Genes Related to Immune Response and Epithelial to Mesenchymal Transition in Explants Lacking Fgfrs

Since several forms of cataract are associated with inflammation and epithelial to mesenchymal transition (EMT), we wished to analyze our data in light of these two important drivers of lens pathology. Given our previous discovery that vitreous exposure elicits a strong innate immune response in wildtype lens epithelial explants, we investigated this response in TKO, QKO and PTEN explants through the course of 10 days of vitreous exposure using the same set of immune genes that we previously showed [24] were induced by vitreous in the wildtype explants (Figure 10A). Wildtype explants samples showed a robust increase in immune response genes through D10, consistent with the earlier finding. QKO samples were closer to wildtype gene-expression patterns for immune genes. At D1, PTEN samples exhibited a significant downregulation of immune genes, with the expression of these genes increasing somewhat at D5 and/or D10. Likewise, at D1, with the exception of *Mmp3*, the immune response genes examined demonstrated only weak expression in the TKO explants, though not as low as in the PTEN explants at D1. The expression of several of these immune genes (*Klhl6*, *Ccl6*, *Spp1*, *Tnfrs11b*, *Mmp9*, *Mmp8*, *Mmp13*, *Mmp12*, *C3*, *Tgfbi*, *Crispld2*, *Fn1* and *Mmp7*) increased in the TKO explants at D5 where they reached their peak expression within this genotype. Overall, the expression of the immune-related genes was suppressed in the TKO relative to the wildtype explants. The QKO explants demonstrated an overall higher expression of immune-related genes than the TKO explants, but lower expression than in the wildtype explants. 

To comprehensively explore the expression of genes related to EMT in either the QKO versus the TKO or in the QKO versus the PTEN explants in response to vitreous-containing DM, we again utilized GSEA, this time relying on the Hallmark dataset with the D10 transcriptomic data. Although EMT is often associated with pathology, when we compared wildtype FVB explants with TKO explants at D10, the wildtype explants showed an EMT enrichment score of 0.38, demonstrating that the loss of FGFRs lowered the EMT response. However, this same analysis comparing the QKO to the TKO or FVB explants resulted in an enrichment score of 0.44 and 0.30 for the QKO sample, respectively. The highest EMT enrichment score was obtained by comparing the QKO to PTEN explants where the enrichment score for the QKO sample was 0.61. This analysis demonstrated that the QKO explants exhibited the highest expression of EMT genes, surpassing the FVB, TKO and PTEN explants (Figure 10B). A selected list of EMT-related genes (*Col5a3*, *Acta2*, *Runx1*, *Runx2* and *Tgfb2*) were tested by RT-qPCR (Appendix A) and were specifically elevated in the QKO genotype relative to all other genotypes (FVB, TKO and PTEN). 

## 4. Discussion

The first demonstration that fiber cell differentiation could be influenced by the ocular media came from the classic experiments by Columbre and Columbre where the reversal of the polarity of the embryonic chick lens within the eye caused the posterior-facing lens epithelial cells to elongate while the peripheral lens epithelium migrated to the anterior to form a new epithelium [56]. Later, McAvoy discovered that the active fiber-differentiation activity within the vitreous could largely be attributed to FGF [57]. However, studies have demonstrated that vitreous humor promotes a stronger differentiation response than purified FGF [58]. Although FGF and vitreous both induce a sustained ERK activation in lens epithelial explants, vitreous induces a stronger activation of AKT than does FGF [1,59]. 

We previously showed in lens epithelial explants that the loss of *Fgfrs* blocked the ability of vitreous humor to induce lens cell elongation and the accumulation of MIP, β-crystallin and γ-crystallin, but that these features were induced by vitreous in explants lacking both *Fgfrs* and *Pten* [25]. These results demonstrated in the absence of PTEN, other factors present in vitreous humor can induce lens fiber cell differentiation. We also recently conducted a transcriptomic analysis of vitreous humor-induced fiber cell differentiation in wildtype lens epithelial explants [24] showing that vitreous humor results in a progressive increase in the expression of many genes associated with fiber cell differentiation and corresponding decrease in the expression of genes associated with lens epithelial cells through five days of culture. We also showed that vitreous-induced gene-expression changes consistent with inflammation and EMT in lens epithelial explants. Here we did a similar study through ten days of culture to determine how the loss of *Fgfrs*, *Pten* or both *Fgfrs* and *Pten* affects vitreous-induced differentiation, inflammation and EMT response. We also sought to determine if a comparison of gene expression in explants lacking all three *Fgfrs* (TKO) with that in explants lacking *Fgfrs* and *Pten* (QKO) could provide insights into which RTK signals were required for vitreous-induced fiber cell differentiation in the absence of FGFR-signaling.

Although the expression of genes associated with lens fiber differentiation (for example: *Hsf4*, *Mip*, *Lim2*, *Gja8*, *Gja3*, *Dnase2b*, *Crybb2*, *Crygs* and *Caprin2*) increased progressively throughout the entire 10-day culture in vitreous humor-containing differentiation media (DM) in wildtype explants, this increase was largely blocked in explants lacking *Fgfrs*. The explants lacking only *Pten* exhibited a pattern of increase in the expression of genes associated with fiber cell differentiation that largely matched that of the wildtype explants, consistent with the relatively mild effect on fiber cell differentiation in vivo in mice lacking *Pten* in the lens [20,60]. Although the pattern of fiber cell-associated gene expression in the QKO explants was lower than that of the wildtype and PTEN explants, these genes were expressed at a significantly higher level than in the TKO explants. 

Vitreous-induced gene-expression changes in wildtype explants results in significant alterations of chromatin structure [24]. The PRC2 repressive complex represents an important epigenetic modifier of gene expression through its ability to induce H3K27 trimethylation (H3K27me3) [61]. Within the lens, the PRC2 component EZH2 has been associated with EMT [62,63] and the prevention of cellular senescence [64]. However, recent studies have suggested that PRC2 may play an important role for suppressing the expression of genes associated with retinal development in the embryonic lens [45]. Our transcriptomic data showed an increase in the expression of many genes associated with PRC2 histone methylation during DM exposure in both the wildtype and PTEN explants. This pattern of PRC2-related gene expression was blocked in the TKO explants and largely restored in the QKO explants. Many of the PRC2-related genes identified in our dataset were also reported in lens fiber cells from the cell atlas of the human ocular anterior segment (Appendix A) confirming that these genes are normal components of the lens transcriptome [44]. 

With respect to genes associated with fiber cell differentiation and PRC2 histone methylation, the pattern of gene expression is most similar in the wildtype and PTEN explants, weakest in the TKO explants and intermediate in the QKO explants. It is possible that FGFR-signaling acts to open the chromatin in the relevant regions to facilitate the expression of these genes while PTEN normally acts to keep these regions in a more closed state. Histone H1 binds to the nucleosome to facilitate higher order chromatin structure associated with transcriptional repression. The C2 domain of PTEN is known to interact with the C-terminal domain of histone H1 to stabilize the association of histone H1 to the nucleosome [65,66]. The loss of PTEN also results in the hyperacetylation of histone H4 and the dissociation of heterochromatin proteins [67]. Therefore, in addition to counteracting FGFR-signaling in the cytoplasm via the dephosphorylation of PIP3 to suppress AKT activation, PTEN can act directly in the nucleus to regulate global gene transcription. FGFRs are also known to go to the nucleus, but what they do there has largely remained a mystery [68]. Furthermore, FGFR2 has been shown to directly phosphorylate PTEN on tyrosine 240 in glioma cells which affects PTEN-chromatin interactions independent of PTEN’s phosphatase activity [69]. Although further experimental evidence will be required to demonstrate direct antagonism between FGFR-signaling and PTEN with respect to chromatin structure, our data suggest this relationship.

While the lens epithelium expresses both FGFRs and PDGFRα, FGFR-signaling has been associated with fiber cell differentiation and PDGFR-signaling associated with lens maintenance of the progenitor pool of cells in lens epithelium [70]. Within the lens, PDGF-signaling induces a stronger AKT activation than does FGF-signaling, while FGFR-signaling induces a stronger and more sustained activation of ERK than is achieved by PDGFR-signaling [1,70]. This difference between signaling outputs from PDGF and FGF might also be explained by the receptor expression. While FGFR expression increases, the expression of PDGFRα declines as fiber cells begin to differentiate [5,6]. While PTEN’s inhibition of AKT activation is well known, PTEN also antagonizes ERK activation as well [71,72]. During embryonic development, the ERK signaling induced by RTKs other than FGFR may be too weak to overcome the suppression by PTEN (as observed in the absence of *Fgfrs* in which *Pten* is still present). However, in the absence of PTEN-based suppression, the ERK activation by the non-FGFR RTKs is sufficient to induce a fiber cell-differentiation response in the absence of *Fgfrs*. Although multiple RTKs may contribute to fiber cell differentiation in the absence of FGFR-signaling, our data suggest that PDGFR signaling plays a relatively more important role than IGF-1R-signaling. This is supported both by GSEA and the increase in PDGFRα expression in lens epithelial explants when *Pten* is deleted. 

The addition of vitreous humor to wildtype lens epithelial explants induces the expression of many genes associated with immune response throughout the 10-day culture period. The lens capsule normally prevents direct exposure of lens cells to either aqueous or vitreous humor. Direct exposure of lens epithelial cells to vitreous humor would suggest that the lens had been injured and the increased expression of immune response genes upon vitreous exposure might be an inherent lens injury response. These same immune response genes are expressed at significantly lower levels, particularly at day 1, in both the TKO and PTEN explants. Although the expression of these genes in the QKO explants does not reach the level seen in the wildtype explants, this genotype gives a response much closer to the wildtype than either the TKO or PTEN explants. 

The induction of EMT in lens epithelial cells is strongly associated with cataract and posterior capsular opacification. In pairwise GSEA comparisons of all four genotypes (FVB, TKO, QKO and PTEN), vitreous exposure induced the lowest EMT response in the PTEN explants and highest response in the QKO explants. Wildtype lens epithelial explants (explants with *Fgfr*s and *Pten* intact) underwent an epithelial–mesenchymal transition (EMT) response when treated with vitreous [24]. Intriguingly, when both *Fgfrs* and *Pten* were absent, the EMT response was notably higher, potentially due to the upregulation of PDGFRα and the lack of PTEN-mediated inhibition. Multiple reports suggest a PDGFR-mediated EMT response [73]. However, lens explants lacking *Pten* surprisingly exhibited a low level of EMT. Although further experiments will be necessary to explain this discrepancy, it is possible that the loss of *Pten* in the lens upregulates microRNAs known to inhibit EMT, such as miR-200 [74,75]. 

## 5. Conclusions

Previously, we conducted a transcriptomic analysis of lenses lacking *Fgfr2*, *Pten* or both *Fgfr2* and *Pten* [23]. However, the cells of the embryonic lens depend on FGFR-signaling for survival [5,17,18,25], making it nearly impossible to study the interplay of FGFR-signaling and PTEN with respect to lens fiber cell differentiation in vivo. Therefore, we adopted the model of lens fiber cell differentiation in vitro using lens epithelial explants induced to differentiate with vitreous humor. This well-established model of lens fiber cell differentiation has been used with explants from chick [76,77], rat [1,28,58,78,79] and mouse [24,25]. In contrast to cells of the embryonic lens, postnatal lens cells do not require FGFR-signaling for survival making it possible to study the effects of FGFR- and PTEN-loss on the fiber-differentiation process without the induction of apoptosis. We previously demonstrated that while loss of FGFR-signaling blocked vitreous-induced lens fiber cell differentiation (as measured by elongation, and the expression of MIP, β-crystallin and γ-crystallin proteins) the loss of both *Fgfrs* and *Pten* restored the ability of vitreous to induce a differentiation response [25]. Here, for the first time, we have conducted a detailed transcriptomic analysis of lens cells induced to differentiate in the absence of *Fgfrs*, *Pten* or both *Fgfrs* and *Pten*. Our findings support that in an *Fgfr*-deficient genetic background, PTEN plays a major role in preventing vitreous-induced fiber cell differentiation. Furthermore, we suggest that signaling from PDGFRα plays an important role in lens fiber cell differentiation when both *Fgfr*s and *Pten* are deleted. Furthermore, we demonstrate that intact FGFR-signaling is required for the initiation of an innate immune response induced by vitreous humor on lens epithelial explants [24]. Finally, we demonstrate that in lens epithelial explants, the loss of either *Pten* or *Fgfrs* reduced features of EMT (relative to wildtype explants) but when both *Pten* and *Fgfrs* are deleted the EMT response surpassed that of wildtype explants. These findings should provide a solid foundation for future investigations on these signaling pathways and their roles in both lens pathology and posterior capsular opacification.

## Figures and Tables

**Figure 1 cells-13-01222-f001:**
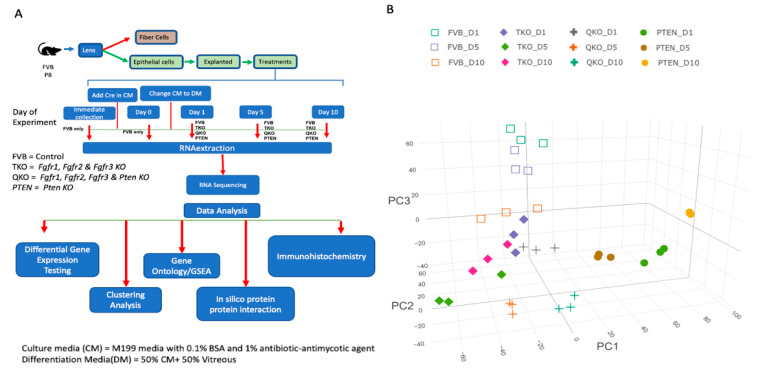
Profiling gene expression in lens epithelial explants. (**A**) Overall experimental procedure. (**B**) A three-dimensional principal component analysis plot shows tight clustering of the three replicates within each group. All FVB (wildtype) samples are represented by diamonds, TKO samples are represented by squares, QKO are represented by crosses and PTEN samples are represented by circles.

**Figure 2 cells-13-01222-f002:**
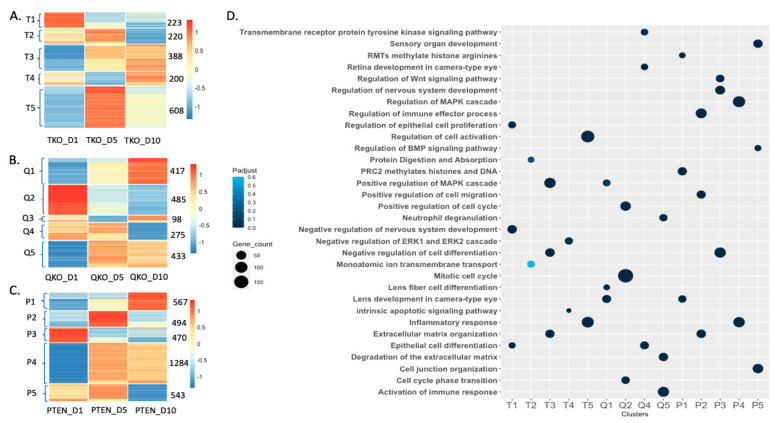
Clustering of differentially expressed genes reveals a distinct signature of the transcriptome as a function of time in differentiation media (DM) within each mutant genotype. K-mean clustering was performed on all DEGs within each genotype across D1 vs. D5, D1 vs. D10 and D5 vs. D10 comparisons by the criteria log_2_ fold change ≥ 1.5 and adjusted *p*-value (*p*-adjust) of ≤0.005, resulting in five clusters for each genotype. (**A**) TKO *n* = 1639 DEGs, (**B**) QKO *n* = 1708 DEGs, (**C**) PTEN *n* = 3358 DEGs. (**D**) Bubble plot represents select gene ontology terms for the clusters identified for all culture conditions. The significance is plotted for each GO term representing the *p*-adjust value.

**Figure 3 cells-13-01222-f003:**
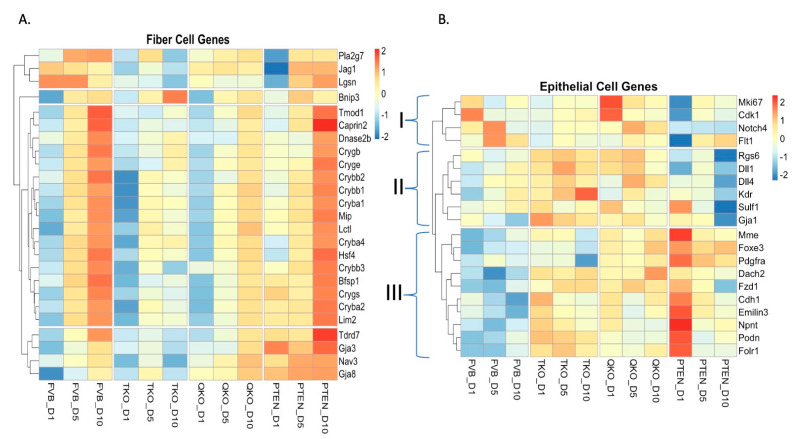
The loss of *Pten* in *Fgfr*-deficient explants partially restored vitreous-induced fiber cell differentiation and induction of PRC2-related mediators of chromatin structure. (**A**) Heatmap indicates z-score adjusted expression values to reveal the expression pattern of fiber cell genes on all conditions and all time points. FVB_D1, FVB_D5 and FVB_D10 samples represent the wildtype lens epithelium explant culture in differentiation media for one day, five days and ten days, respectively, after 24 h in culture media. TKO_D1, TKO_D5 and TKO_D10 represent the (*Fgfr1-3)* knockout samples cultured in differentiation media for one day, five days and ten days, respectively, after 24 h in culture media. QKO_D1, QKO_D5 and QKO_D10 represent the (*Fgfr1-3* and *Pten*) knockout samples cultured in differentiation media for one day, five days and ten days, respectively, after 24 h in culture media. PTEN_D1, PTEN_D5 and PTEN_D10 represent the (*Pten*) knockout samples cultured in differentiation media for one day, five days and ten days, respectively, after 24 h in culture media. (**B**) Heatmap indicates z-score adjusted expression values to reveal the expression pattern of epithelial cell genes on all conditions and all time points as described for part (**A**).

**Figure 4 cells-13-01222-f004:**
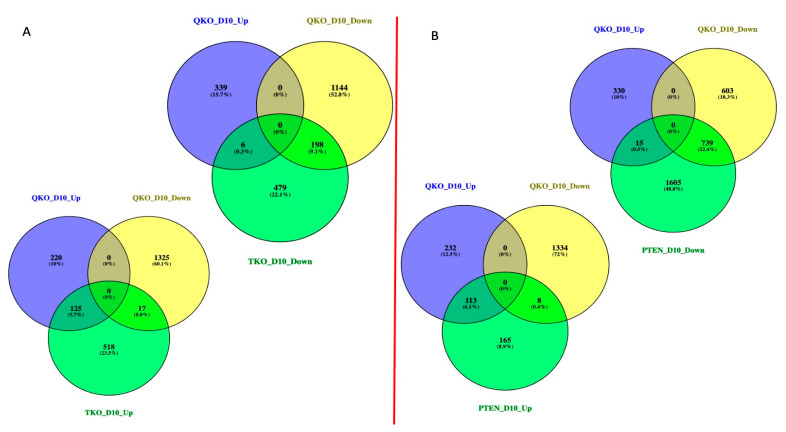
Analysis of differentially expressed genes on the D10 stage for TKO, QKO and PTEN to identify epistatic relationships. (**A**) Venn diagram displays the genes differentially expressed in TKO and QKO at D10 stage. (**B**) Venn diagram displays the genes differentially expressed in PTEN and QKO at D10 stage.

**Figure 5 cells-13-01222-f005:**
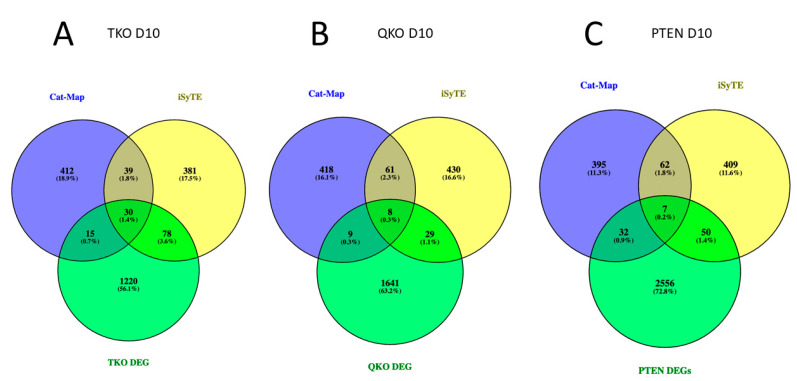
Venn diagram displays the differentially expressed genes in TKO, QKO and PTEN at D10 stage intersected with iSyTE database and Cat-Map database.

**Figure 6 cells-13-01222-f006:**
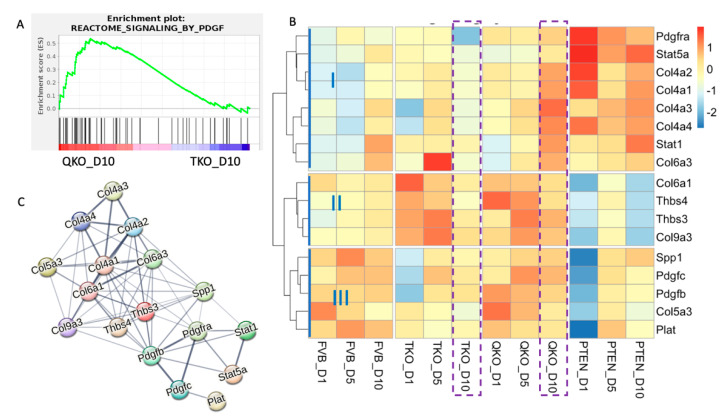
QKO_D10 samples showed enrichment for PDGF signaling. (**A**) Gene set enrichment analysis (GSEA) was performed on normalized RNA-seq-expression values from QKO_D10 and TKO_D10 samples. Genes are ordered along the *x*-axis based on expression rank between the two conditions. Black bars indicate genes associated with a given term. The green line indicates the enrichment score determined by GSEA. (**B**) Heatmap indicates z-score adjusted expression values to reveal the expression pattern of PDGF-signaling genes on all conditions and all time points. Purple box represents the expression for D10 samples in TKO and QKO. (**C**) STRING network visualizing functional interactions (edges) between proteins (nodes) significantly enriched in the explant samples. The thickness of the edges corresponds to their score, and the default STRING clustering confidence score cutoff of 0.4 was used to determine whether two nodes were functionally related.

**Figure 7 cells-13-01222-f007:**
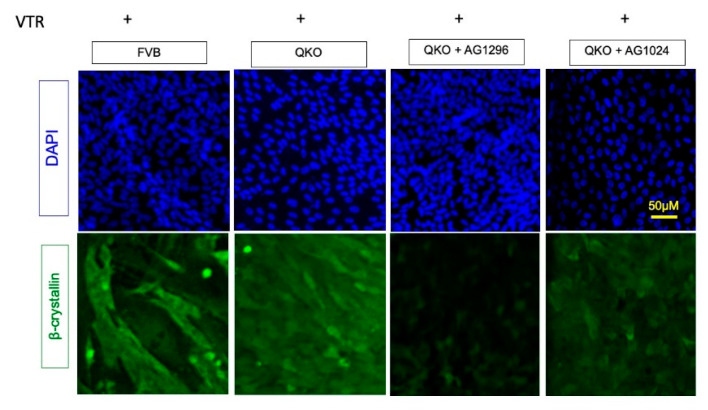
PDGFRa inhibitors block the β-crystallin expression in QKO explants. Immunohistochemistry analysis of FVB and QKO explants for 5 days in DM shows the β-crystalline expression. Addition of 20 μM AG1296 (PDGFRa inhibitor) 2 h before the media change and in DM for 5 days blocks the robust β-crystalline expression in QKO however addition of IGF receptor (IGF-1R) inhibitor (AG1024) 5 μM failed to block the induction of β-crystallin in QKO explants.

**Figure 8 cells-13-01222-f008:**
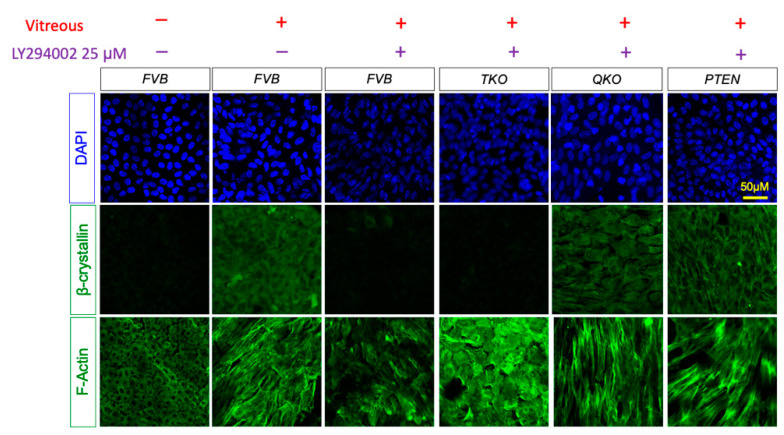
Immunohistochemistry analysis for vitreous-induced fiber cell-differentiation response in *FVB*, TKO (*Fgfrs* 1–3 knocked out), QKO (*Fgfrs* 1–3 and *Pten* knocked out samples), PTEN (*Pten* knocked out samples) explants in presence of PI3K inhibitor LY294002. Immunohistological stain for the fiber cell marker β-crystallin and phalloidin stain to indicate F-actin arrangement on FVB, TKO, QKO and PTEN explants following 5 days in vitreous and the selective PI3K inhibitor LY294002.

**Figure 9 cells-13-01222-f009:**
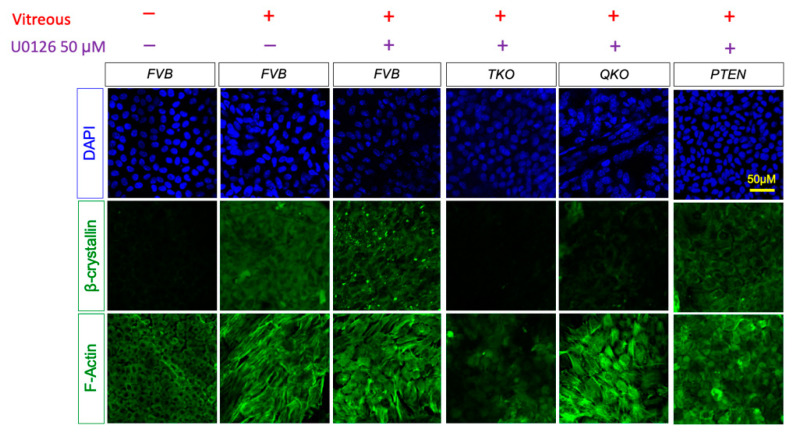
Immunohistochemistry analysis for vitreous-induced fiber cell-differentiation response in *FVB*, TKO (*Fgfrs* 1–3 knocked out), QKO (*Fgfrs* 1–3 and *Pten* knocked out samples), PTEN (*Pten* knocked out samples) explants in presence of MEK inhibitor U0126. Immunohistological stain for the fiber cell marker β- crystallin and phalloidin stain to indicate F-actin arrangement on FVB, TKO, QKO and PTEN explants following 5 days in vitreous and the selective MEK inhibitor U0126.

**Figure 10 cells-13-01222-f010:**
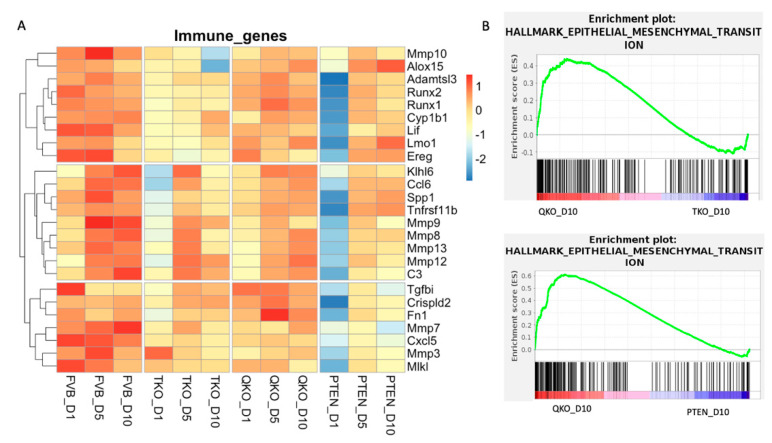
Immune response and epithelial to mesenchymal transition were significantly enriched in QKO samples. (**A**) Heatmap indicates z-score adjusted expression values to reveal expression patterns of immune-related genes identified from our previously published manuscript for all conditions and all time points. (**B**) Gene set enrichment analysis (GSEA) was performed on normalized RNA-seq-expression values from QKO_D10, TKO_D10 and PTEN_D10 samples. Genes are ordered along the *x*-axis based on expression rank between the two conditions. Black bars indicate genes associated with a given term. The green line indicates the enrichment score determined by GSEA.

**Table 1 cells-13-01222-t001:** Pdgfra is reduced in several gene-knockout mouse models with lens defects. The iSyTE database [47] was examined for Pdgfra expression in several gene-specific knockout mouse models that exhibit lens defects. iSyTE contains meta-analysis of microarray datasets isolated from lens tissue of Pax6cKO (Le-Cre:Pax6loxP:loxP lens conditional KO; stages embryonic day (E) 9.5, E10.5 [53]), E2F1/2/3cKO (MLR10-Cre:E2F1-/-:E2F2-/-:E2F3fl/fl lens conditional KO; E17.5 and postnatal day (P) 0 (newborn)) [54], Notch2cKO (Le-Cre:Notch2loxP:loxP lens conditional KO; E19.5 [55]). Lens conditional knockout of Pax6 E2F1/E2F2/E2F3 and Notch2 result in significant reduction (*p* < 0.05) of *Pdgfra* transcripts in lens tissue.

Mouse Model	*Pax6^cKO^*	*Pax6^cKO^*	*E2F1:2:3^cKO^*	*E2F1:2:3^cKO^*	*Notch2^cKO^*
Stage	E9.5	E10.5	E17.5	P0	E19.5
Pdgfra (f.c.)	−1.6	−6.6	−1.9	−6.4	−4.2

## Data Availability

The data discussed in this publication have been deposited in NCBI’s Gene Expression Omnibus (Edgar et al., 2002: https://doi.org/10.1093/nar/30.1.207) and are accessible through GEO Series accession number GSE264405. All code will be made available upon reasonable request to the authors.

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
