# Peer review of "A Transcriptomics Analysis of the Regulation of Lens Fiber Cell Differentiation in the Absence of FGFRs and PTEN"

_cells, 2024, doi:10.3390/cells13141222_

Round 1
Reviewer 1 Report
Comments and Suggestions for Authors
This manuscript by Upreti et al., describes a comprehensive analysis of fiber cell differentiation in the absence of FGFRs and PTEN in mouse lens epithelial explants. The requirement for growth factor signaling, in particular FGFRs, for lens fiber cell differentiation has long been established. The downstream opposing signals have only recently been elucidated and this manuscript builds on that knowledge to examine the transcriptome of explants in the presence or absence of FGFRs and the opposing signal PTEN. Here the authors use vitreous as a source of FGF which has been found induce fiber cell differentiation in conjunction with other growth factors contained within. Vitreous induced sustained ERK1/2 signaling via receptor tyrosine kinases in lens explants to promote fiber cell differentiation is a well-established model for fiber cell differentiation.
Previous work by the authors showed that loss of all three FGFrs resulted in loss of fiber cell differentiation and that this loss could be partially rescued by knockout of PTEN. This was achieved for the most part by examining key fiber cell specific markers. The data presented here is a novel analysis of the gene expression changed resulting from the loss of growth factor signaling or its opposing signal.
Here the authors used explants that were ultimately knocked out for all three FGFrs, FGFr and opposing signal PTEN and PTEN alone, and as control wild type FVB explants. RNA was collected on D1, D5 and D10 following addition of the differentiating media (DM), media with 50% vitreous). The authors then used RNAseq to examine the transcriptome of three genotypes over the 10-day period, the RNAseq analysis is thorough, appropriate and clearly presented. Analysis of transcript levels of genes normally expressed in fiber cells confirmed the differing gene expression patterns resulting from the loss of FGF and/or PTEN signaling.
Overall the findings confirm previous data regarding the opposing FGFr and PTEN interaction in terms of fiber cell gene expression as well as novel findings on chromatin remodeling. They also uncover an important upregulation of PDGFRa signaling in the absence of FGFrs and PTEN as well as demonstrating the importance of ERK signaling in the absence of FGFrs and PTEN.
The analysis is sound and conclusions are supported by the data presented. Only minor points are listed to be addressed.
1. The heatmap of epithelial cell genes in supplementary figure 2 might be better as part of figure 3 as it shows a good contrast to the upregulation of fiber cell genes.
2. Line 396 contains a typo – it should say 79 genes instead of 38
3. Figure 4 is very hard to make out.
4. It is a little unclear as to why there is an immune response in the lens epithelial explant following addition of vitreous. At Day 1 in the FVB the immune modulator transcript levels are very high, do the authors think this is the result of trauma to the lens or directly a result of vitreous addition. The vitreous is of Bovine origin could that be an issue? Although the gene expression results are discussed along with the EMT marker increases it’s not clear why there would be an immune response perhaps the authors could explain why they think this is happening and why it’s important.
Author Response
Reviewer 1:
- The heatmap of epithelial cell genes in supplementary figure 2 might be better as part of figure 3 as it shows a good contrast to the upregulation of fiber cell genes.
As the reviewer suggested, we have included the heatmap of epithelial cell genes to figure 3 as part B. of figure 3, We have also moved the PRC2 methylates histone heatmap (originally 3B) to a new supplemental figure 2.
- Line 396 contains a typo – it should say 79 genes instead of 38
We were a little confused by this comment as there was no mention of gene number in line 396. However, we did discuss 38 genes in line 386. In this case, we were indeed analyzing the expression of 38 genes, rather than 79 genes, because only half of the 79 genes listed under the gene ontology term “PRC2 methylates histones and DNA”, were differentially expressed in the PTEN explants. To clarify the language in this section, we revised the portion of the paragraph describing figure 3B (which is now figure S2) as follows:
Therefore, we surveyed all 79 of the genes identified by the gene ontology program Metascape using the GO term "PRC2 methylates histones and DNA". Half of these (38 genes) were differentially expressed in the PTEN explants, and the expression of these genes was analyzed across all explant genotypes (FVB, TKO, QKO and PTEN) for all timepoints (Figure S2).
- Figure 4 is very hard to make out.
We were uncertain of the reviewer’s meaning here. We responded by increasing the size of the figure and increasing the size of the fonts to make the figure easier to see.
- It is a little unclear as to why there is an immune response in the lens epithelial explant following addition of vitreous. At Day 1 in the FVB the immune modulator transcript levels are very high, do the authors think this is the result of trauma to the lens or directly a result of vitreous addition. The vitreous is of Bovine origin could that be an issue? Although the gene expression results are discussed along with the EMT marker increases it’s not clear why there would be an immune response perhaps the authors could explain why they think this is happening and why it’s important.
We were initially surprised that the exposure of vitreous humor induced such a strong immune response. We do not believe that this response results from the use of bovine vitreous humor as opposed to vitreous humor from another species as the lens explants should be free from most of the cell types (B- and T-lymphocytes) responsible for adaptive immunity. We also do not believe that the trauma of explanting explains this result because in our previous manuscript (Upreti et al 2023) we did not find such a strong immune response in explants cultured without the addition of vitreous humor. Instead, we hypothesize that lens epithelial cells respond to direct exposure of vitreous humor as an injury response. In vivo, lens epithelial cells would not be exposed directly to vitreous humor in the absence of traumatic injury. To emphasize this point, we added the following to the discussion:
The lens capsule normally prevents direct exposure of lens cells to either aqueous or vitreous humor. Direct exposure of lens epithelial cells to vitreous humor would suggest that the lens had been injured and the increased expression of immune response genes upon vitreous exposure might be an inherent lens injury response.
Reviewer 2 Report
Comments and Suggestions for Authors
In this manuscript, authors used a lens epithelial explant system and conducted a transcriptomic analysis of lens cells induced to differentiate in the absence of Fgfrs, Pten, or both Fgfrs and Pten. When both Pten and Fgfrs are missing, PDGFRα is important for lens differentiation, and the EMT response has undergone significant changes. The study is interesting and valuable for understanding the role of lens pathology and posterior capsular opacification.
There are some comments for the authors.
1. Please provide more experiments to further demonstrate the important role of PDGFRand ERK signaling in absence of FGFRs and PTEN. The authors may try detecting the expression of β-crystallin in lens explants using qRT-PCR or western blotting in figure 7 and 8.
2. Please supplement the experiment to verify the gene expression level of EMT in QKO, FVB, TKO, and PTEN explants.
3. The scale bar in Figure 7 is missing.
4. References are generally old, please increase the proportion of references from the last 5 years.
Comments on the Quality of English LanguageThe English language in your manuscript would benefit from improvement for clarity and readability.
Author Response
Reviewer 2:
- Please provide more experiments to further demonstrate the important role of PDGFR and ERK signaling in absence of FGFRs and PTEN. The authors may try detecting the expression of β-crystallin in lens explants using qRT-PCR or western blotting in figure 7 and 8.
To address the reviewer comments, we have added three supplemental figures and a supplemental table that provide more data for the roles of PDGFR, ERK and PI3K in the absence of FGFRs and PTEN. Figure S6 is an immunofluorescent study using an antibody to MIP. This shows that MIP staining is evident in vitreous exposed explants from FVB (control) and in QKO explants. Our previous work shows the MIP staining is lost in the TKO explants (Padula et al., 2020). We show that the MIP signal is lost in the QKO explants when cultured in the presence of the PDGFR inhibitor AG1296, but only partially reduced in QKO explants cultured in the presence of the IGFR inhibitor AG1024.
Figure S7 is an RT-qPCR analysis of Crybb2 and Crybb3 gene expression in FVB, and QKO (with and without the PDGFR inhibitor AG1296 or the IGFR inhibitor AG1024). This figure shows that the highest expression of these β-crystallins is in the FVB explant with the QKO and QKO cultures with AG1024 having nearly equivalent levels of expression and the lowest expression seen in the QKO explants cultured in AG1296. Again, figure S7 demonstrates the relatively greater importance of PDGFR-signaling than IGFR-signaling for maintaining fiber-cell differentiation responses in the QKO explants. This result is entirely consistent with and supports figure 7.
Figure S8 is another RT-qPCR analysis of Crybb2 and Crybb3 gene expression in the FVB (control), TKO, QKO and PTEN explants cultured in the presence of the PI3K inhibitor LY294002. The expression of both of these genes is very low in the FVB and TKO explants, but rises significantly in the QKO and PTEN explants, both of which are deleted for Pten. This result is entirely consistent with and supports figure 8.
Figure S9 is an RT-qPCR analysis of Crybb2 and Crybb3 gene expression in the FVB (control), TKO, QKO and PTEN explants cultured in the presence of the MEK inhibitor U0126. The expression of these genes is reasonably similar in the FVB and PTEN explants, but markedly reduced in the TKO and QKO explants. This result is entirely consistent with and supports figure 9.
In aggregate, figures 8, S8, 9 and S9 find that β-crystallin expression is more dependent on the PI3K arm of FGFR-signaling than the ERK-arm of FGFR-signaling. This result is consistent with that previously found by Wang Q, Stump R, McAvoy JW, Lovicu FJ. MAPK/ERK1/2 and PI3-kinase signalling pathways are required for vitreous-induced lens fibre cell differentiation. Exp Eye Res. 2009 Feb;88(2):293-306 (Figure 5) and Lovicu FJ, McAvoy JW. FGF-induced lens cell proliferation and differentiation is dependent on MAPK (ERK1/2) signalling. Development. 2001 Dec;128(24):5075-84 (Figure 9).
We added section 2.8 to the materials and methods to describe how the RT-qPCR analyses were done:
2.8. RT-qPCR
To validate RNA-Seq/Immunofluorescence results, selected genes were analyzed using RT-qPCR. RNA extraction followed the previously described protocol, and samples were reverse transcribed into cDNA using oligo(dT) and random primers, following the ImProm-IITM Reverse Transcription System (A3800) manufacturer's instructions. RT-qPCR assays were conducted with three biological replicates and three technical replicates using GoTaq qPCR Master Mix (A6001) and the Bio-Rad CFX Connect. Pre-designed primers were sourced from PrimerBank (https://pga.mgh.harvard.edu/primerbank/, accessed on 16 May 2024) and obtained from IDT Technologies (Table S19)
The comparative ∆∆Ct method was used to determine relative gene expression levels compared to housekeeping genes (Gapdh) and relative mRNA was normalized to FVB. To determine the statistical significance of the observed differences in gene expression, we performed an analysis of variance (ANOVA) on the logFC values, considering the effects of treatment and gene. This was followed by Tukey's Honest Significant Difference (HSD) post-hoc test to identify specific group differences. The results of the Tukey HSD test were used to annotate the plots with significance levels, indicated as 'ns' for non-significant, * for p < 0.05, ** for p < 0.01, and *** for p < 0.001.
Table S19 was added to describe all the primers used in RT-qPCR analyses.
- Please supplement the experiment to verify the gene expression level of EMT in QKO, FVB, TKO, and PTEN explants.
We added a supplemental figure (S10) to further validate the increased expression of EMT-related genes (Col5a3, Acta2, Runx1, Runx2, and Tgfb2) in the QKO genotype relative to the other genotypes. We added the following text to the results section immediately before the discussion:
A selected list of EMT-related genes (Col5a3, Acta2, Runx1, Runx2, and Tgfb2) were tested by RT-qPCR (Figure S10) and were specifically elevated in the QKO genotype relative to all other genotypes (FVB, TKO and PTEN).
- The scale bar in Figure 7 is missing.
The scale bar is now included on Figure 7.
- References are generally old, please increase the proportion of references from the last 5 years.
We respectfully disagree with this reviewer’s comment. The references included in the manuscript are the most relevant rather than most recent references. Also the general practice is to reference what source discovered a particular result first rather than most recently. To arbitrarily include more recent references which are not the most original sources of the referenced information would be less, rather than more appropriate for this manuscript. If the reviewer has specific, more recent references that should be included that we have mistakenly or carelessly omitted we will be happy to reconsider.
The English language in your manuscript would benefit from improvement for clarity and readability.
We are left confused as to what the reviewer found unclear or difficult to read. We were given no specific guidance as to how to improve the readability or clarity of the manuscript. We did revise one statement as described under reviewer 1’s comment #2 were we felt that we were not as clear as we needed to be. We also revised several other portions of the manuscript with the revisions found in the revised manuscript Microsoft word file indicated in red text. Six of the eight authors on this manuscript are native English speakers and all have contributed to the final manuscript.